# Specific Glucagon Assay System Using a Receptor-Derived Glucagon-Binding Peptide Probe

**DOI:** 10.3390/ijms27010515

**Published:** 2026-01-04

**Authors:** Hajime Shigeto, Yoshio Suzuki, Shohei Yamamura

**Affiliations:** 1Health and Medical Research Institute, National Institute of Advanced Industrial Science and Technology (AIST), 2217-14 Hayashi-cho, Takamatsu 761-0395, Japan; 2Health and Medical Research Institute, National Institute of Advanced Industrial Science and Technology (AIST), Central 6, 1-1-1 Higashi, Tsukuba 305-8566, Japan; suzuki-yoshio@aist.go.jp

**Keywords:** glucagon, glucagon receptor, biosensor

## Abstract

Glucagon is a peptide hormone secreted by pancreatic alpha cells which elevates blood glucose and plays a critical role in diabetes onset and homeostasis. The accurate assessment of glucagon concentration is challenging due to its structural similarity with other hormones, causing cross-reactivity in antibody-based methods. Rapid and specific glucagon detection is essential, particularly during hypoglycemia. This study aimed to develop glucagon-specific probes combining high specificity, rapid detection, and ease of operation. We designed novel peptide-based probes by screening glucagon-binding peptides from the glucagon receptor sequence using a peptide array method. This strategy, based on receptor amino acid sequences, can be applied to the identification of binding peptides for other hormones, expanding its potential utility. The screened peptides were conjugated with fluorescent dyes to create probes enabling detection within 30 min. The developed probes demonstrated superior specificity for glucagon relative to similar sequence analogs compared with conventional antibody-based methods, with detection limits in the nanomolar range. This study represents a proof-of-concept approach for rapid and highly specific glucagon detection. However, further optimization of probe sensitivity and validation under physiological conditions will be required before clinical or diagnostic application. These improvements in the probe’s properties will enable the reliable blood glucagon detection and accurate diagnostic assessment of diabetes-related diseases.

## 1. Introduction

Glucagon is a peptide hormone secreted by pancreatic alpha cells (α-cells) in response to low blood glucose levels. Glucagon induces the degradation of glycogen accumulated in the cells into glucose to increase blood glucose levels [1,2,3,4,5,6,7]. Mouse model studies have shown that both the inhibition of glucagon signaling and insulin secretion does not result in the diabetes phenotype [8,9,10]. These results indicate that glucagon contributes more to the maintenance of blood glucose concentrations than insulin. Additionally, in individuals with type 1 diabetes, α-cell dysfunction leads to impaired glucagon secretion, compromising the counter-regulatory response to low glucose levels and increasing the risk of severe hypoglycemic events [11]. The precise and rapid quantification of circulating glucagon is therefore critical for optimizing glycemic control and mitigating acute complications. However, the significance of glucagon was not established until recently because of difficulty of its accurate detection using conventional methods. Glucagon is matured from proglucagon through enzymatic cleavage. Various peptides with similar amino acid sequences are produced from proglucagon depending on their cleavage sites [12,13]. Therefore, antibody-based detection methods (e.g., enzyme-linked immunosorbent assay [ELISA]) often detect false positive signals from reactions involving proglucagon-derived peptides, with certain analogs exhibiting cross-reactivity of approximately 30%. Since peptides derived from proglucagon include oxyntomodulin, glicentin, and glucagon-like peptide-1 (GLP-1), some of which contribute to the reduction in blood glucose levels, detection methods for these peptides must have high specificity and accuracy [13,14,15,16,17,18]. Therefore, to accurately detect peptides derived from proglucagon as a diagnostic marker, liquid chromatography–mass spectrometry (LC-MS) must be used; however, it difficult to perform in regular clinics. Therefore, conventional antibody-based methods like ELISA are the candidate method to detect glucagon; however, higher specificity is needed for blood samples. It is especially reported that the concentration of glucagon in the basal human plasma is at a low level, almost 50 pM [7,14]. Recently, the glucagon-specific antibodies that bind to the N- or C-terminus of glucagon have been identified and can be used for glucagon detection using sandwich ELISA methods [19,20,21]. Although the ELISA methods enable the detection of the target peptide with high sensitivity and specificity, the methods are time-consuming and require the technical handling of complicated procedures, typically requiring more than 16 h to ensure specific antibody binding. This prolonged processing time makes them unsuitable for point-of-care applications, which are essential for responding to sudden hypoglycemic events. Although in silico approaches for designing glucagon binders have also been reported [22], achieving high specificity de novo remains extremely challenging due to the structural similarity of proglucagon-derived peptides. Therefore, a novel, highly accurate, rapid, and easy operation method needs to be developed. Glucagon receptors are one of the G-protein-coupled receptors (GPCRs). Glucagon binds to the extracellular domain (ECD) and transmembrane domain region, which is expressed from the cellular membrane [23]. Some probes have been developed using sequences or functional domains from the target receptor [24,25], and several peptide-based probes have been reported that showed potential for the highly sensitive and specific analysis of various biomarkers, such as caspase-3, glypican-3, human angiotensin-converting enzyme 2 (hACE2) for SARS-CoV-2, and GLP-1 receptor [26,27,28,29]. This demonstrates that leveraging receptor-derived sequences provides an effective strategy for developing probes targeting a wide range of peptides and proteins. Glucagon receptors exhibit specificity for the target peptide [30,31,32]; therefore, peptides from glucagon receptors may also have glucagon-specific binding ability. However, it is necessary to screen peptide sequences with easy handling and high specificity from the receptor sequence for the development of the peptide-based assay method. In this study, we evaluated glucagon-binding peptides (GBPs) from the entire receptor sequence to develop a novel antibody-independent glucagon-detecting probe. We developed novel GBPs by conjugating the fluorescent dye that increases fluorescence intensity through its response to target molecules. The use of short peptides enables cost-effective probe synthesis, and when combined with fluorescence-based detection, this approach provides a simple and rapid measurement method. The functional analysis of GBPs was examined by evaluating its detection sensitivity (Kd value) and specificity compared with ELISA methods. The assay methods developed in this research demonstrated comparable or higher glucagon specificity compared to conventional ELISA methods. However, the detection sensitivity was still lower. These results indicate that, for measuring the physiological and pathological levels of circulating glucagon, more sensitive and improved probes need to be developed in the future. However, the findings on the probe developed in this study would facilitate the development of a simple method for diagnosing blood glucagon levels with high specificity.

## 2. Results and Discussion

### 2.1. Development of GBP by Isolating Peptides from Glucagon Receptor Sequences

To develop the novel GBP with high binding affinity and specificity, glucagon-binding peptides were screened (Figure 1). Several organs express the glucagon receptor on the cellular membrane, which react specifically to glucagon. Therefore, more sensitive and specific GBPs may be able to develop from receptor sequences. The peptide-based probes, which consisted of short peptides, were easy to synthesize chemically to have high usability. The GBP sequence from glucagon receptors was screened from the peptide array [33,34], which consisted of 94 types of 15 lengths of amino acid peptide spots (Appendix A). Cy3-modified glucagon or oxyntomodulin (100 nM), an analog whose N-terminus has the same amino acid sequence as glucagon, was added to the peptide array. After the reaction, the specific binding sequence for glucagon was screened. The fluorescence images showed some areas with strong fluorescent intensities, which indicated the peptides with a binding affinity for glucagon or oxyntomodulin on the receptor sequence (Figure 2a,b). The fluorescence intensities were calculated from each image (Figure 2c,d). Since peptides No. 3 and 59 showed the highest fluorescence intensities, they were presumed to have the highest binding affinity to glucagon. To obtain highly specific peptides, we screened the peptides that had the ability to bind to glucagon but not to oxyntomodulin. Significant differences in fluorescence intensity were observed between glucagon and oxyntomodulin for peptides No. 15, 32, and 65. In contrast, peptide No. 3 corresponded to an N-terminal membrane translocation signal sequence, which is characterized by a high proportion of hydrophobic amino acids. The observed interaction with glucagon is likely attributable to non-specific hydrophobic interactions rather than true glucagon-specific binding. Furthermore, the absence of a defined recognition motif and the potential for non-specific adsorption suggest that this peptide lacks functional relevance for selective glucagon detection. Therefore, peptide No. 3 was excluded from further consideration as a probe candidate. Peptides No. 15, 32, 59, and 65 were selected as candidates for glucagon-binding peptides with high sensitivity and specificity.

To examine the glucagon-binding ability of the selected peptides, an intermolecular interaction analysis was performed using the localized surface plasmon resonance (LSPR) system (Nicoya, ON, Canada). Biotinylated peptides at the N-terminus were synthesized, and the binding affinity to glucagon or other proglucagon-derived peptides (oxyntomodulin, mini-glucagon, and glicentin) were analyzed (Figure 3 and Appendix A). Signal intensities were increased according to glucagon concentration. In this assay, peptides were immobilized at the sensor surface by the biotin–streptavidin system. The peptides No. 15 and 65 showed a lower affinity to glucagon, which may have been caused by biotinylation at the N-terminus. However, the screened peptides have a binding affinity to glucagon. The calculated Kd values of these peptides were 1.15 µM (No. 15), 0.59 µM (No. 32), 0.58 µM (No. 59), and 1.86 µM (No. 65), respectively. Although these Kd values are not as sensitive as those of anti-glucagon antibodies, they are sufficient to detect changes in glucagon concentration within the pM–nM range. However, considering that physiological glucagon levels in vivo are approximately 50 pM, the sensitivity in this study may not be adequate for accurate detection under normal physiological conditions. One possible reason for the relatively low sensitivity is that the measurements were performed in sodium acetate buffer (pH 5.2) to solubilize glucagon, which is a non-physiological condition. Under such conditions, the binding efficiency may have been compromised. In vivo, glucagon exists at much lower concentrations (~50 pM) and under higher pH conditions, closer to physiological levels. Therefore, improving the probe’s performance under more physiologically relevant pH conditions could allow for more sensitive measurements and enhance the applicability of the method.

The results from the LSPR assay indicate that the glucagon-binding domains are located in the ECD (No. 15), transmembrane domain (No. 32 and 59), and tyrosine kinase domain (No. 65) of the receptor. Previous reports showed that glucagon binds to the ECD and some extracellular sites of the transmembrane domain of the glucagon receptor [23]. The glucagon-binding site identified in this study was confirmed to be consistent with previous reports, especially in the No. 15, 32, and 59 peptides. Furthermore, we successfully identified a glucagon-binding peptide derived from a previously unreported domain (No. 65).

This finding highlights the utility of peptide array-based screening using receptor sequences as a powerful approach for discovering specific binding peptides, particularly for targets where conventional antibody-based methods often fail to achieve sufficient specificity.

### 2.2. Functional Analysis of Screened Peptides as GBPs

We modified the screened peptides with fluorescence dyes to develop the GBPs. The amine-reactive fluorescent dye was screened for GBP development. BODIPY558/568 NHS Ester (succinimidyl ester) (Thermo Fisher Scientific, Waltham, MA, USA), sulforhodamine 101 acid chloride (5-[dimethylamino] naphthalene-1-sulfonyl chloride) (Dojindo, Kumamoto, Japan), and fluorescein isothiocyanate isomer I (Dojindo) were tested in this study. Each dye was modified in the screened peptides, and Sulforhodamine 101 acid chloride was selected as the most suitable dye for the GBP. The fluorescence dye-modified peptides (No. 15, 32, 59, and 65) showed strong fluorescence intensity at 600 nm when they bound to glucagon (Figure 4a), and intensities increased with the glucagon concentration. Therefore, it is confirmed that the fluorescent dye-modified peptides functioned as glucagon-detecting probes. These changes were caused by a complex formation, in which the fluorophore of the fluorescent peptide is bound at the hydrophobic positions of glucagon supported by the hydrophobic interaction. This interaction undergoes the intramolecular charge transfer process, the quantum yields of which linearly respond to the amount of glucagon in solution [35,36]. The detection limit of the probes was 5 nM. The glucagon concentration [7,10] required for the diagnosis of diabetes is 20–50 pM; thus, the GBPs may need to improve in sensitivity for diagnostic applications in future. To evaluate the potential diagnostic applicability of GBPs, glucagon-spiked human plasma samples were analyzed using the developed probes. All four GBPs exhibited increased signals in the spiked samples compared to the non-spiked plasma (Appendix A), suggesting that the GBP-based assay could be adapted for diagnostic use. However, the signal intensity was significantly reduced compared to measurements in pure buffer, indicating possible interference from plasma components. Although GBPs were not affected by BSA, human plasma contains various proteins, lipids, and metabolites that may compete for binding sites or alter probe conformation, thereby reducing effective interaction with glucagon. Importantly, these results demonstrate that high concentrations of glucagon in human plasma can be successfully detected using the developed probes. This suggests that, with further improvements in probe sensitivity, detection at low physiological concentrations of glucagon may also be achievable. Furthermore, because the native glucagon receptor is capable of strong binding to glucagon at physiological levels due to its 3D conformation, studying the structural optimization of the GBPs will enable clinically relevant measurements, paving the way for its application in diagnostic settings.

In addition, although the GBPs have comparable or higher specificity compared to conventional ELISA methods and are simpler, making them more practical than conventional methods, the GBPs showed lower measurement stability, thereby requiring further improvements to the peptides. For example, more sensitive and stable GBPs may be able to obtain by the peptide array analysis using the random mutation-induced peptide sequence. In this research, a short sequence of 15 amino acids is used; therefore, it is easy to make a peptide array in which mutations have been introduced into whole amino acid positions.

The specificity of the GBP was established by measuring the proglucagon-derived peptide concentrations (i.e., oxyntomodulin, mini-glucagon, glicentin, and GLP-1). These peptides were matured through the cleavage from proglucagon. Furthermore, a part of their amino acid sequences were the same as those of glucagon. This similarity in amino acid sequences makes glucagon-specific detection difficult using antibody-based detection methods like ELISA. The GBPs developed in this study reacted with each peptide, and their fluorescence intensities were measured. Figure 4b shows that the GBPs had a lower affinity with other proglucagon-derived peptides than with glucagon. Especially for oxyntomodulin, approximately 10% more signals were detected than for glucagon signals. These results indicate that the GBPs had a low affinity for oxyntomodulin. Glucagon and oxyntomodulin have the same amino acid sequence at the N-terminal. In contrast, mini-glucagon, which has the same sequence as glucagon at the C-terminal, did not show an increase in fluorescence intensity. These results indicate that the GBPs bind to the N-terminus of glucagon. Additionally, the cross-reactivity (glucagon vs. other proglucagon-derived peptides) of GBPs were compared with that of ELISA. The same samples were measured by both GBP and ELISA, and the signal intensities obtained were compared. The GBP showed 0–20% reactivity to the glucagon analog, while ELISA showed 30% reactivity (Figure 4b). The results confirmed that the GBP has a higher specificity for mini-glucagon and oxyntomodulin and the same level for glicentin and GLP-1 compared to the commercial ELISA system using the same sample and protocol. Furthermore, the GBP-based assay offers a significant practical advantage over conventional ELISA, which typically requires more than 16 h to complete. In contrast, the GBP method achieves high specificity with only a simple and easy operative fluorescence measurement after a short incubation period of approximately 30 min. If the detection sensitivity can be further improved, this approach could be fully applicable in clinical contexts such as diabetes—particularly in cases where the rapid and precise measurement of glucagon is critical, such as hypoglycemia—making the GBP-based detection method a highly efficient and reliable alternative.

To examine the binding site of GBP on glucagon, we performed a peptide array analysis using sequences derived from peptides No. 15, 32, 59, and 65, with systematic deletions from the N- or C-terminus. This approach allowed us to identify regions critical for glucagon recognition. In particular, peptide No. 15, which lacked three amino acids at the C-terminus, exhibited a marked reduction in glucagon-binding affinity (Figure 5), suggesting that the C-terminal region of GBP plays an essential role in interacting with the N-terminus of glucagon. In contrast, peptide No. 32 showed increased fluorescence intensity upon C-terminal deletion, indicating that its C-terminal segment may sterically interfere with glucagon binding. These functional observations are consistent with the hypothetical structural predictions generated by AlphaFold2 (Figure 1b and Appendix A), which position the C-terminal residues of GBP within the predicted glucagon-binding interface. These results provide structural insights into the determinants of GBP–glucagon interaction and suggest that the rational modification of the GBP sequence could further enhance binding affinity. Improving sensitivity and specificity based on these structural findings may ultimately enable the development of a rapid, simple, and highly specific glucagon assay suitable for clinical applications.

For the fluorescent images of the peptide array reacted with Cy3-modified glucagon, the sequences of peptides No. 15, 32, 59, and 65 that were deleted from the N- or C-terminus were synthesized on the array. The peptide array was excited with a 532 nm laser, and images were captured using a 3D-Scanner. Data are expressed as mean ± standard deviation of three replicates.

## 3. Materials and Methods

### 3.1. Peptide Array Synthesis

A peptide array of 15 amino acid peptides was constructed by overlapping 10 amino acids along a sequence (Appendix A) of glucagon receptors. The array is synthesized on cellulose membranes (grade 542; Whatman, Maidstone, UK) activated with β-alanine as the N-terminus using a peptide auto-spotter (MultiPep Rsi, Intavis AG, Köln, Germany), as previously described [31,32]. For the addition of each amino acid, the synthesis cycle was initiated by deprotecting the Fmoc-protecting group with 20% piperidine in N,N-dimethylformamide (DMF) before washing the membrane with DMF and ethanol. Prior to amino acid coupling, Fmoc amino acids at 0.5 M were activated by 1.1 M hydroxybenzotriazole and 1.1 M N,N-diisopropylcarbodiimide. After the coupling step, the remaining unreacted amino groups were blocked with 4% acetic anhydride in DMF and subsequently washed with DMF and ethanol. The synthesis was conducted according to the manufacturer’s instructions with some modifications. After the final cycle, the Fmoc- and side-chain-protecting groups were manually removed with 20% piperidine in DMF and 2% acetic anhydride in a mixture of DMF and Milli-Q water, triisopropyl silane, and trifluoroacetic acid (2:3:95). Finally, the membrane was thoroughly washed with dichloromethane, DMF, ethanol, and phosphate-buffered saline (PBS).

### 3.2. Screening the Glucagon-Binding Peptide from Peptide Array

The peptide array was blocked with 1% bovine serum albumin (BSA) containing 50 mM Tris-buffered saline (TBS). The blocked peptide array was washed with the washing buffer, 0.05% Tween 20 containing 50 mM TBS. Cy3-modified 100 nM glucagon or oxyntomodulin (Peptide Institute, Osaka, Japan) was added and reacted for 1 h at room temperature. The fluorescence images were obtained using a 3D-Gene Scanner (Toray, Tokyo, Japan) after washing the array with the washing buffer. The fluorescence intensity of each spot was calculated using the Image J ver. 1.54 software.

### 3.3. Intermolecular Interaction Analysis of Synthesized Peptides

The binding affinity of the screened peptides with the peptide array was elucidated by localized surface plasmon resonance (LSPR) analysis. Glucagon-binding peptides screened by peptide array were purchased from the Peptide Institute. The synthesized peptides were biotinylated using a protein biotinylating kit (Thermo Fisher Scientific, Waltham, MA, USA) at the N-terminus of each peptide, according to the manufacturer’s protocol. The binding affinity of the biotinylated peptides against glucagon or other proglucagon-derived peptides were measured with an Alto digital LSPR system (Nicoya, Kitchener, ON, Canada) using a streptavidin surfacing kit according to the manufacturer’s protocol. Briefly, the sensor of the Alto carboxyl cartridge was first immobilized with streptavidin and reacted with the biotinylated peptide (10 μM) diluted in 100 mM sodium acetate buffer (pH 5.2; Nacalai Tesque, Kyoto, Japan) containing 0.1% Tween 20 (Sigma Aldrich, St. Louis, MO, USA). Then, 10 or 100 μM glucagon (Peptide Institute), oxyntomodulin (Peptide Institute), mini-glucagon (Fujifilm Wako Pure Chemical, Osaka, Japan), or glicentin (Fujifilm Wako Pure Chemical) were reacted after dilution at 3, 9, 27, 81, or 243 times in 100 mM sodium acetate buffer (pH 5.2) containing 0.1% Tween 20. Signal responses were measured according to the manufacturer’s protocol. The Kd values were calculated using the built-in software of the Alto system ver. 2.2.1.

### 3.4. GBP Synthesis by Modifying the Fluorescent Dye in Glucagon-Binding Peptides

The synthesized peptides were dissolved in triethylamine-added MQ or dimethyl sulfoxide (DMSO). The fluorescent dye sulforhodamine 101 acid chloride (5-[dimethylamino] naphthalene-1-sulfonyl chloride) (Dojindo, Kumamoto, Japan) was dissolved in DMSO and modified at the N-terminus of each peptide by reacting with the solutions for 2 h at room temperature and 16 h at 4 °C. The fluorescent dye-modified peptides were dried and washed twice with toluene using a Rotavapor R-100 vacuum distiller (Buchi Labortechnik AG, Flawil, Switzerland). The unmodified fluorescent dye was dissolved with dichloromethane and the supernatant was removed. After the fluorescent dye-modified peptides were again dried by a Rotavapor R-100 vacuum distiller (BÜCHI Labortechnik AG, Flawil, Switzerland), the peptides were dissolved in DMSO to prepare the GBP stock solution.

### 3.5. Glucagon Assay Using the GBP

Functional analysis of the screened peptides as GBPs was measured by fluorescent analysis with glucagon and other proglucagon-derived peptides. Oxyntomodulin, mini-glucagon, glicentin, and GLP-1 (7–36) (Peptide Institute) were used as the other proglucagon-derived peptides in this study. Each peptide was dissolved in PBS or 0.01% acetic acid solution and reacted with the fluorescent dye-modified peptide (10 µM). The 600 nm fluorescent intensity excited at 568 nm was measured with a Hitachi F-7000 fluorescent spectrometer (Hitachi, Tokyo, Japan). The intensities were normalized using the intensity of each solution without the target glucagon or the other proglucagon-derived peptides. To compare the sensitivity and specificity between the GBP-based and ELISA-based assays, the glucagon and analog were measured using a glucagon ELISA kit (Mercodia, Uppsala, Sweden) according to the manufacturer’s protocol. The results were normalized as described above.

### 3.6. Limitations and Future Works

In this stage, there are some issues which remain in this study. First, the current detection limit (5 nM) of the GBP-based assay is not sufficient to reliably measure physiological glucagon concentrations (~50 pM) in human plasma. Second, stability and reproducibility in plasma were not evaluated, which are critical for clinical applicability. Thus, future work will focus on improving the probe’s sensitivity to enable detection at lower concentrations, assessing assay stability under physiologically relevant conditions, and further enhancing specificity to minimize cross-reactivity. Additionally, biological activity testing—such as receptor activation assays or cell-based functional studies—will be incorporated to validate the physiological relevance of the detected glucagon. These steps, along with validation using clinical samples and reproducibility testing, will be essential for establishing clinical applicability.

## 4. Conclusions

This study developed a novel glucagon-binding peptide (GBP) for the highly specific and rapid detection of glucagon. Using a novel screening system with glucagon receptor-derived sequences and a peptide array, four GBPs were successfully conjugated with fluorescent dyes, enabling rapid detection within 30 min under easy operational conditions. Compared to the conventional ELISA, the developed probes exhibited higher specificity for the similar amino acid sequence peptides. These findings demonstrate the high potential of peptide array-based screening for developing novel peptide-based probes from receptor sequences. This work represents a proof-of-concept study demonstrating the feasibility of a receptor-derived peptide screening strategy for rapid and highly specific glucagon detection. Although the current assay sensitivity is insufficient for physiological glucagon concentrations, further improvements in probe affinity, assay conditions, and validation under physiologically relevant settings will be required before clinical or diagnostic applications can be considered. These advancements in the GBP assay may significantly enhance the diagnostic evaluation and management of diabetes-related disorders.

## Figures and Tables

**Figure 1 ijms-27-00515-f001:**
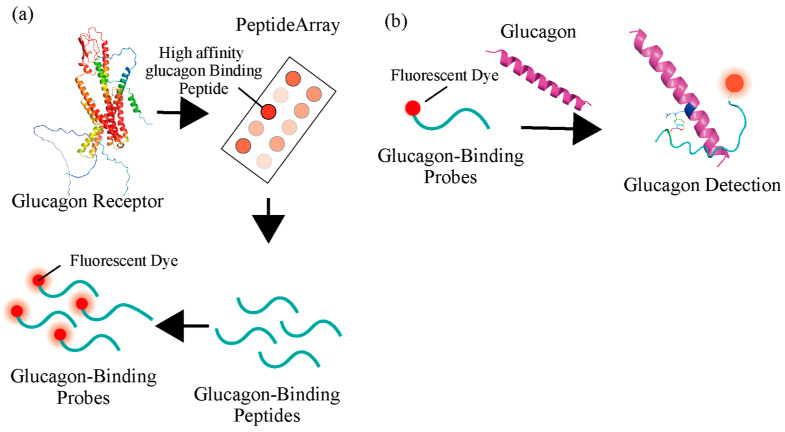
Schematic concept of the development of the glucagon-binding probe (GBP). (**a**) Development of the GBP. Peptides with glucagon-binding affinity were screened from the glucagon receptor using a peptide array. The screened peptides were modified with fluorescent dye. The structure of the glucagon receptor was illustrated using AlphaFold2 ver. 1.5.5 https://colab.research.google.com/github/sokrypton/ColabFold/blob/main/AlphaFold2.ipynb (accessed on 1 May 2024). (**b**) Principle of target glucagon detection with GBP. The binding mode of glucagon and peptide No. 15 was estimated using AlphaFold2 ver. 1.5.5 and is presented as a hypothetical model. This prediction is not experimentally validated and should be interpreted as an illustrative hypothesis rather than confirmed structural data.

**Figure 2 ijms-27-00515-f002:**
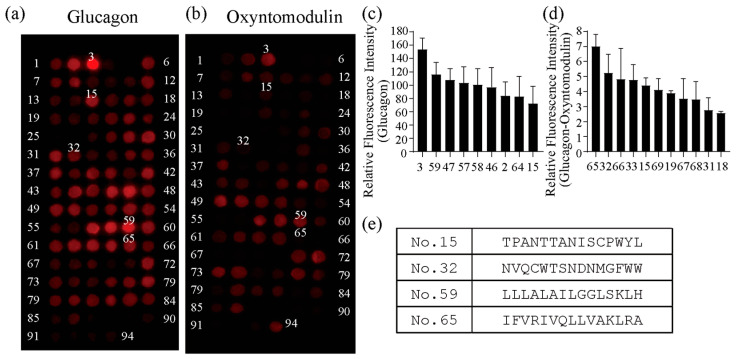
Screening for glucagon-binding peptides of the glucagon receptor sequence by peptide array analysis (**a**,**b**). Fluorescent images of peptide arrays reacted with Cy3-modified (**a**) glucagon or (**b**) oxyntomodulin. The 94 types of 15 amino acid peptides were synthesized on the array. The peptide array was excited with a 532 nm laser. (**c**) Fluorescence intensities of each spot emitted strong fluorescence in (**a**). Data are expressed as mean ± standard deviation of three replicates. (**d**) Relative fluorescence intensities of each spot normalized by the oxyntomodulin-reacted intensity in (**b**). Data are expressed as mean ± standard deviation of three replicates. (**e**) The screened peptide sequences as glucagon-binding peptide candidates.

**Figure 3 ijms-27-00515-f003:**
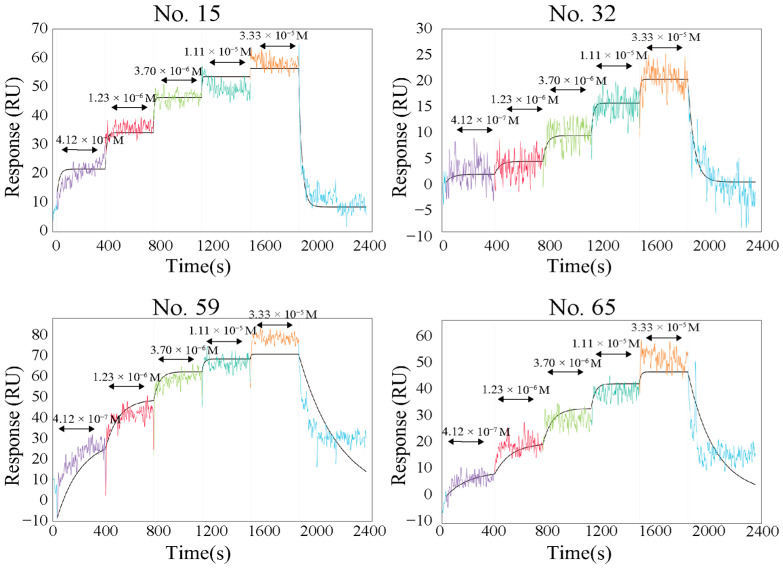
Intramolecular functional analysis using a localized surface plasmon resonance system. Intramolecular functional analysis of screened peptides (No. 15, 32, 59, and 65) and glucagon. Each biotinylated peptide was immobilized at the sensor’s surface at the reaction cartridge. Glucagon (10 or 100 μM) was gradually and automatically diluted with sodium acetate buffer (pH 5.2) containing 0.1% Tween 20 and reacted to immobilized peptides. The bidirectional arrows indicate the glucagon concentration reacted against immobilized peptides.

**Figure 4 ijms-27-00515-f004:**
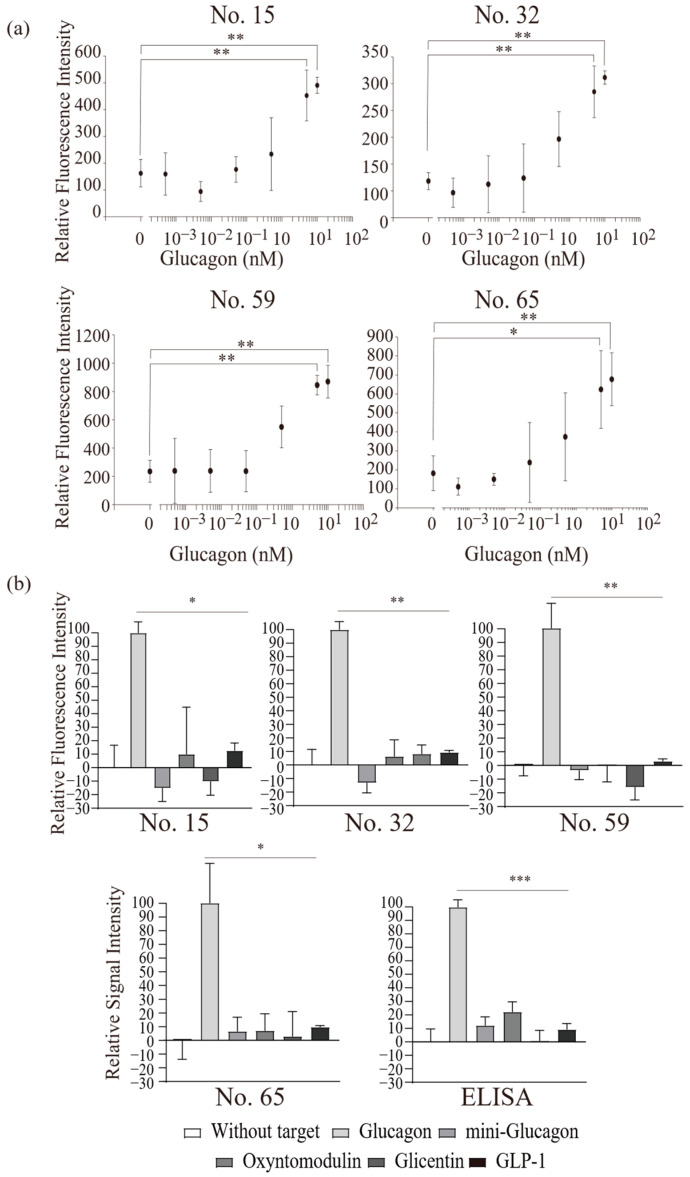
Functional analysis of glucagon-binding probes (GBPs) for glucagon measurement. The GBPs were reacted with glucagon or the other proglucagon-derived peptides. (**a**) Concentration-dependent responses of the fluorescent intensities for each GBP. Data are expressed as mean ± standard deviation of three to five replicates. * *p* < 0.05, ** *p* < 0.01, one-way ANOVA followed by Tukey’s comparison test. [The one-way ANOVA revealed significant differences among groups for all tested conditions. No. 15; F(6, 14) = 12.38, *p* < 0.001; Tukey tests: 0 nM vs. 5 nM (*p* = 0.002), 0 nM vs. 10 nM (*p* = 0.002). No. 32; F(6, 14) = 13.29, *p* < 0.001; Tukey tests: 0 nM vs. 5 nM (*p* = 0.002), 0 nM vs. 10 nM (*p* = 0.001). No. 59; F(6, 14) = 12.81, *p* < 0.001; Tukey tests: 0 nM vs. 5 nM (*p* = 0.002), 0 nM vs. 10 nM (*p* = 0.001). No. 65; F(6, 14) = 6.84, *p* < 0.002; Tukey tests: 0 nM vs. 5 nM (*p* = 0.015), 0 nM vs. 10 nM (*p* = 0.007).] (**b**) Measurement of cross-reactivity against proglucagon-derived peptides using the GBPs and enzyme-linked immunosorbent assay methods. Relative intensities were normalized by the intensities without target peptides and glucagon. Data are expressed as mean ± standard deviation of three to five replicates. * *p* < 0.05, ** *p* < 0.01, *** *p* < 0.001, one-way ANOVA followed by Tukey’s comparison test. [The one-way ANOVA revealed significant differences among groups for all tested conditions. For No. 15, F(5, 27) = 22.41, *p* < 0.001; Tukey tests: glucagon vs. mini-glucagon (*p* = 0.001), vs. oxyntomodulin (*p* = 0.013), vs. glicentin (*p* < 0.001), vs. GLP-1 (*p* = 0.002). For No. 32, F(5, 24) = 72.52, *p* < 0.001; Tukey tests: glucagon vs. mini-glucagon (*p* = 0.001), vs. oxyntomodulin (*p* = 0.001), vs. glicentin (*p* = 0.001), vs. GLP-1 (*p* = 0.004). For No. 59, F(5, 30) = 74.73, *p* < 0.001; Tukey tests: glucagon vs. mini-glucagon (*p* = 0.007), vs. oxyntomodulin (*p* = 0.004), vs. glicentin (*p* = 0.004), vs. GLP-1 (*p* = 0.009). For No. 65, F(5, 25) = 28.36, *p* < 0.001; Tukey tests: glucagon vs. mini-glucagon (*p* = 0.018), vs. oxyntomodulin (*p* = 0.015), vs. glicentin (*p* = 0.008), vs. GLP-1 (*p* = 0.025). For ELISA, F(5, 22) = 6.925, *p* < 0.001; Tukey tests: glucagon vs. mini-glucagon (*p* < 0.001), vs. oxyntomodulin (*p* < 0.001), vs. glicentin (*p* < 0.001), vs. GLP-1 (*p* < 0.001)].

**Figure 5 ijms-27-00515-f005:**
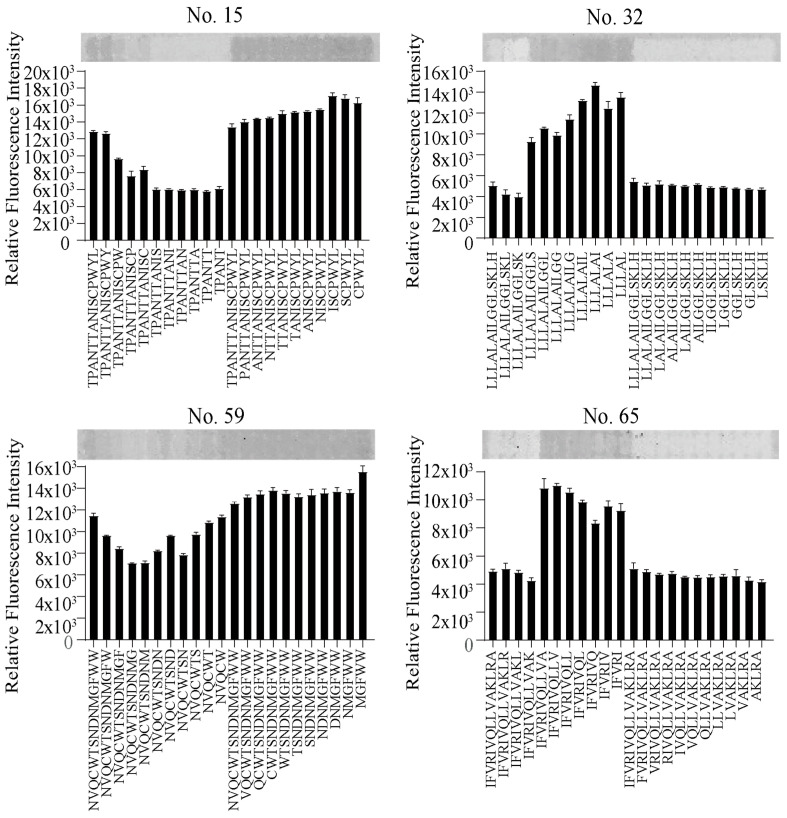
Elucidating the glucagon-binding site of the glucagon-binding probe.

## Data Availability

The original contributions presented in this study are included in the article/Appendix A. Further inquiries can be directed to the corresponding authors.

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
