# Peer review of "Specific Glucagon Assay System Using a Receptor-Derived Glucagon-Binding Peptide Probe"

_ijms, 2026, doi:10.3390/ijms27010515_

Round 1

Reviewer 1 Report

Comments and Suggestions for Authors

Reviewer Report

Manuscript Title: Specific Glucagon Assay System Using Receptor-Derived Glucagon-Binding Peptide Probe
Manuscript ID: IJMS-4036468

Dear Authors,

I would like to begin by sincerely thanking you for the opportunity to review your manuscript. Your work addresses an important analytical challenge in the field of glucagon biology, and the idea of using receptor-derived glucagon-binding peptides (GBPs) as an alternative to antibody-based assays is innovative and promising. The manuscript demonstrates significant experimental effort, and the peptide array screening strategy is particularly valuable for generating novel peptide probes.

My comments below aim to help strengthen the scientific clarity, structure, and impact of your manuscript so that its contribution can be fully appreciated by the scientific community.

  1. Abstract

Comments

  • The abstract should clearly articulate:
    • the existing gaps in glucagon detection (cross-reactivity, low specificity, slow assays),
    • the aim of developing a GBP-based probe,
    • the novelty and innovation of the receptor-derived approach,
    • and key performance outcomes (specificity, detection time, limitations).
  • Please revise the abstract so it is structured, gap-driven, and highlights how your method improves over ELISA.

  1. Introduction

Comments

  • The introduction is informative but overly contains historical information that:
  •  May be condensed to keep space to add:
  • Current challenges in glucagon detection should be presented more explicitly:
    • cross-reactivity with proglucagon-derived peptides,
    • low specificity of antibody kits,
    • technical complexity and time of ELISA,
    • the need for rapid point-of-care tests,
    • cost-effectiveness considerations.
  • Clearly define the gap, the aim, and the specific objectives.
  • Please explain why a receptor-derived GBP represents an innovative alternative to conventional methods.

  1. Results

3.1 Peptide Screening (Figures 1–2)

  • Figures require higher resolution and clearer labeling for easy interpretation.
  • Consider marking high-affinity peptides directly on the images.
  • Please provide more detail on why peptide No.3 was excluded (hydrophobic non-specific binding).

3.2 LSPR Binding Studies (Figure 3)

  • The Kd values achieved are in the micromolar range, which is much higher than physiological glucagon concentrations (≈50 pM). This limitation should be acknowledged clearly.
  • Buffer pH (5.2) is non-physiological; please discuss whether this may influence binding affinity.

3.3 Fluorescent GBP Performance (Figure 4)

  • The work nicely demonstrates the improved specificity over ELISA.
  • However:
    • More discussion is needed on human plasma matrix interference.
    • No stability or reproducibility data are presented or discussed, if you have not , then state them in limitation section in order to open next research
    • Please quantify the practical advantages (30-min workflow, simple mixing, ease of use).
  • Authors should clearly state that sensitivity needs further enhancement before clinical application.

3.4 Deletion Analysis (Figure 5)

  • The peptide deletion mapping is valuable but needs clearer explanation linking structure to binding.
  • Please improve figure clarity and align findings with AlphaFold predictions.

  1. Discussion

Comments

  • The Discussion section reiterates results; it should instead:
    • interpret findings in depth,
    • explain the advantages of GBP over ELISA,
    • acknowledge limitations:
      • current detection limit (5 nM),
      • micromolar Kd values,
      • limited plasma validation,
      • no clinical samples,
      • no reproducibility data,
    • include practical considerations (cost-effectiveness, point-of-care potential).
  • A more structured discussion will improve clarity.

  1. Conclusion

Comments

  • Please add a clear future perspective, including:
    • peptide sequence optimization,
    • directed evolution or mutagenesis,
    • dual-probe detection systems,
    • clinical sample validation,
    • eventual point-of-care device development.

  1. Figures and Presentation

Comments

  • Several figures need improved resolution and clearer axis labeling.
  • English language throughout the manuscript requires editing for clarity and consistency.
  • Please ensure all abbreviations (e.g., GBP) are introduced fully at first mention.

  1. References

Comments

  • The reference list does not include any citations from 2024 or 2025 except one in 2024. to ensure the study reflects the most recent advancements in the field, the authors should consider incorporating more up-to-date references.
  • Please ensure:
    • key peptide-based biosensor literature is included,
    • ELISA specificity and variability papers are cited

  1. Overall Assessment

Strengths

  • Innovative receptor-derived peptide concept.
  • Strong potential for superior specificity.
  • Rapid (30-minute), mix-and-read workflow.
  • Valuable peptide array screening methodology.

Major Limitations

  • Sensitivity currently insufficient for clinical glucagon levels.
  • Limited validation in plasma.
  • Figures need improvement.
  • Abstract, introduction, and discussion need restructuring.

Author Response

Dear Reviewer 1,

Thank you very much for your kind and careful review of our manuscript entitled “Specific glucagon assay system using receptor-derived glucagon binding peptide probe.”
We have revised the manuscript according to your comments and suggestions, and have addressed all points one by one as detailed in the attached file.
Please refer to the attached file for detailed responses and revisions.
We believe that our revisions and responses meet the requirements for publication in the Special Issue “Molecular Research on Proglucagon-Derived Peptides” of the International Journal of Molecular Sciences.
We appreciate your consideration and look forward to your feedback.

Sincerely yours,

Hajime Shigeto

Reviewer 2 Report

Comments and Suggestions for Authors

The manuscript presents a peptide-based glucagon detection strategy by screening glucagon receptor–derived sequences and developing fluorescent peptide probes (GBPs). The concept is scientifically interesting, and such antibody-independent detection could make a meaningful contribution to diagnostic development. The study is generally well structured; however, several issues related to clarity, methodology,  and data presentation should be addressed before acceptance.

Here are my comments

- Sensitivity remains insufficient for physiological glucagon levels: The functional assay indicates a detection limit of ~5 nM, whereas physiological fasting plasma glucagon is ~20–50 pM. The authors acknowledge this limitation, but the discussion does not adequately explain:

1- Why is a probe with micromolar Kd values expected to detect picomolar glucagon?

2- Whether the assay can realistically be improved to the necessary range.

3- Whether the presented method has any feasibility for in vivo or clinical plasma detection without significant enhancement.

I highly recommend that the authors expand the discussion to address the gap between current probe sensitivity and clinically relevant concentrations.

- Some Statistical Analysis Requires Strengthening: Most figures simply report mean ± SD of n=3 without clear statistical tests or validation. For example, ANOVA results are mentioned, but F-values, df, and exact p-values are missing.

- Although novelty is the major point of strength in this manuscript, this novelty needs more clarification: More discussion is needed to highlight the novelty versus prior receptor-based probe approaches and peptide-array-based binder discovery.

-  Figures should include scale bars, clearer labels, and higher resolution.

- The manuscript uses AlphaFold2 for predicting binding interactions between glucagon and peptides. AlphaFold2 is not validated for peptide-peptide or peptide-hormone interactions without templates. Authors should clarify that these models are hypotheses rather than validated structural interactions or use docking or MD simulations to strengthen the claim.

- The reference section showed no self-citation; however, some references were outdated

- English needs to be revised as some typos were detected

The study is interesting, and the methodology is promising, but the results as currently presented are not sufficient to support the strong claims of assay specificity and feasibility for clinical application. If the authors address the concerns above, the revised manuscript may be suitable for publication.

Comments on the Quality of English Language

English needs to be revised as some typos were detected 

Author Response

Dear Reviewer 2,

Thank you very much for your kind and careful review of our manuscript entitled “Specific glucagon assay system using receptor-derived glucagon binding peptide probe.”
We have revised the manuscript according to your comments and suggestions, and have addressed all points one by one as detailed in the attached file.
Please refer to the attached file for detailed responses and revisions.
We believe that our revisions and responses meet the requirements for publication in the Special Issue “Molecular Research on Proglucagon-Derived Peptides” of the International Journal of Molecular Sciences.
We appreciate your consideration and look forward to your feedback.

Sincerely yours,

Hajime Shigeto

Reviewer 3 Report

Comments and Suggestions for Authors

The reviewed manuscript is well written, interesting and presents important results.

The aim of the study was to develop a simple method for diagnosing blood glucagon levels with high specificity. It is difficult to measure the plasma levels of glucagon what makes problems in patients with hyperglycemia. For diagnosis of diabetic status it would be extremely useful to measure insulin and glucagon levels, so specific, fast and relatively cheap methods are necessary. Authors developed and described interesting method which enabled to measure specific glucagon fragments hydrolyzed from proglucagon. However, it needs further developing of more sensitive probes.

Comments:

  1. line 49 and others: the term "glucagon analogs" is not adequate. The hormones are not glucagon analogs but physiologically active proglucagon fragments, I am suggesting to change this term "analog".
  2. conclusions -it is rather Summary. I suggest to divide into two separate parts: Summary and Conclusion .

In summary: The results obtained in presented study are interesting and optimistic, well discussed. The M&M section is logically and clearly described. The cited scientific publications were well chosen and confirmed the right methods and results. I would suggest to check the biological activity of measured glucagon in addition to spiked human plasma.

Author Response

Dear Reviewer 3,

Thank you very much for your kind and careful review of our manuscript entitled “Specific glucagon assay system using receptor-derived glucagon binding peptide probe.”
We have revised the manuscript according to your comments and suggestions, and have addressed all points one by one as detailed in the attached file.
Please refer to the attached file for detailed responses and revisions.
We believe that our revisions and responses meet the requirements for publication in the Special Issue “Molecular Research on Proglucagon-Derived Peptides” of the International Journal of Molecular Sciences.
We appreciate your consideration and look forward to your feedback.

Sincerely yours,

Hajime Shigeto

Round 2

Reviewer 1 Report

Comments and Suggestions for Authors

Dear Authors,

The revised manuscript has substantially improved and appropriately addresses the reviewer’s previous comments. The authors now clearly and transparently discuss the technical limitations of the assay, including sensitivity, non-physiological assay conditions, and the need for further validation.

To further strengthen the manuscript and avoid any potential overinterpretation, I recommend explicitly stating in both the Abstract and Conclusion that the present work represents a proof-of-concept study, and that additional optimization (e.g., sensitivity improvement to physiological glucagon levels and validation under physiological conditions) is required before clinical or diagnostic application.

This clarification is already well supported by the Results and the Limitations and Future Works section, and making it explicit in the Abstract and Conclusion would improve clarity, accuracy, ethics and editorial robustness without detracting from the novelty or scientific value of the study. 

To help you:

1. You can insert this text at the end of the Abstract:

This study represents a proof-of-concept approach for rapid and highly specific glucagon detection; however, further optimization of probe sensitivity and validation under physiological conditions will be required before clinical or diagnostic application. 

2. You can insert this text as replacement or addition near the end of Section 5:

This work represents a proof-of-concept study demonstrating the feasibility of a receptor-derived peptide screening strategy for rapid and highly specific glucagon detection. Although the current assay sensitivity is insufficient for physiological glucagon concentrations, further improvements in probe affinity, assay conditions, and validation under physiologically relevant settings will be required before clinical or diagnostic applications can be considered.

Author Response

(The authors gave the same response as above.)
